complexity/applied mathematics/mathematical finance

complex systems, financial markets, sentiment analysis, machine learning, transfer entropy

**Authors for correspondence:**
Nino Antulov-Fantulin
e-mail: anino@ethz.ch
Petter N. Kolm
e-mail: petter.kolm@nyu.edu

# On the impact of publicly available news and information transfer to financial markets

Metod Jazbec[1], Barna Pàsztor[1], Felix Faltings[1], Nino Antulov-Fantulin[2,3] and Petter N. Kolm[3]

[1]Department of Computer Science, ETH Zurich, 8092 Zurich, Switzerland
[2]Computational Social Science, ETH Zurich, 8092 Zurich, Switzerland
[3]Courant Institute of Mathematical Sciences, New York University, New York, NY 10012, USA

NA-F, 0000-0002-4337-2475

We quantify the propagation and absorption of large-scale publicly available news articles from the World Wide Web to financial markets. To extract publicly available information, we use the news archives from the Common Crawl, a non-profit organization that crawls a large part of the web. We develop a processing pipeline to identify news articles associated with the constituent companies in the S&P 500 index, an equity market index that measures the stock performance of US companies. Using machine learning techniques, we extract sentiment scores from the Common Crawl News data and employ tools from information theory to quantify the information transfer from public news articles to the US stock market. Furthermore, we analyse and quantify the economic significance of the news-based information with a simple sentiment-based portfolio trading strategy. Our findings provide support for that information in publicly available news on the World Wide Web has a statistically and economically significant impact on events in financial markets.

## 1. Introduction

Studies of the impact of speculation and information arrival on the price dynamics of financial securities have a long history, going back to the early work of Bachelier [1] in 1900 and Mandelbrot [2] in 1963 (see, Jarrow & Protter [3] for a historical account of these and related developments). In 1970, Fama [4] formulated the efficient market hypothesis in financial economics, stating that security prices reflect all publicly available information. Shortly after in 1973, Clark [5] proposed

the mixture of distribution hypothesis which asserts that the dynamics of price returns are governed by the information flow available to traders. Subsequently, novel models were introduced such as the sequential information arrival model [6], news jump dynamics [7], both in the 1980s, and truncated Levy processes from econophysics [8] in the 1990s, to name a few examples. With the rise of the World Wide Web and social media came an ever-increasing abundance of available data, allowing for more detailed studies of the impact of news on financial markets at different time-scales [9–17]. Big data, coupled together with advancements in machine learning (ML) [18] and complex systems research [19–22], enabled more efficient analysis of financial data [23], such as web data [24–39], social media [40–44], web search queries [45–47], online blogs [48,49] and other alternative data sources [50].

In this article, we use news articles from the Common Crawl News, a subset of the Common Crawl's petabytes of publicly available World Wide Web archives, to measure the impact of the arrival of new information about the constituent stocks in the S&P 500 index at the time of publishing. To the best of our knowledge, our study is the first one to use the Common Crawl News data in this way. We develop a cloud-based processing pipeline that identifies news articles in the dataset that are related to the companies in the S&P 500. As the Common Crawl public data archives are getting bigger, they are opening doors for many real-world 'data-hungry' applications such as transformers models such as GPT [51] and BERT [52], a recent class of deep learning language models. We believe that public sources of news data are important not only in natural language processing (NLP) and financial markets, but also in complex systems and computational social sciences that are aiming to characterize (mis)information propagation and dynamics in techno-socio-economic systems. The abundance of high-frequency data in financial markets enables complex systems researchers to have microscopic observables, allowing for the testing and verification of hypotheses and theories previously not possible.

Using ML methods from NLP [53,54] we analyse and extract sentiment for each news article in the Common Crawl News repository, assigning a score in the range from zero to one that represent most negative (zero) through most positive (one) sentiment. To quantify the information propagation from the publicly available news articles on the World Wide Web to companies in the S&P 500 index, we use two different approaches. First, we employ tools from information theory of complex systems [20,22,55] to measure the impact of information transfer of the news sentiment scores on the returns of the constituent companies in the S&P 500 index at an intraday level. Second, we implement and simulate the daily portfolio returns resulting from a simple trading strategy based on extracted news sentiment scores for each company. We use the returns from this strategy as an econometric instrument and compare it with several benchmark strategies that do not incorporate news sentiments. Our findings provide support for that information in publicly available news on the World Wide Web has a statistically and economically significant impact on events in financial markets.

# 2. Methods

## 2.1. Common Crawl data

News data are freely and publicly available from many online sources including newspapers, news outlets and news aggregators. Assembling a dataset that covers a representative subset of these sources is a major task. For example, it took about three and a half years for Google to develop a release-ready version of Google News.[1] In this article, we use a large public dattset from the Common Crawl, a non-profit organization that collects data from the World Wide Web. Since its start in 2008, the Common Crawl has collected petabytes of data and their developers have continued to improve their system and expand the number of websites visited by their crawler.

We performed our analysis using the Common Crawl News, a subset of the Common Crawl dataset containing only news articles.[2] From its start in 2016 until the end of February 2020, Common Crawl collected about eight terabytes of news from publicly available news sources on the World Wide Web, corresponding to a total of about 400 million articles covering a wide range of topics in different languages.

Figure 1 depicts the steps we took in processing the Common Crawl News dataset. Processing this dataset is complicated due to its size, the content's raw format, the different HTML formats used on websites and the broad coverage of different topics and languages. Furthermore, as part of the

---

[1]See https://googleblog.blogspot.com/2006/01/and-now-news.html.

[2]This version of the dataset is available at https://commoncrawl.org/2016/10/news-dataset-available.

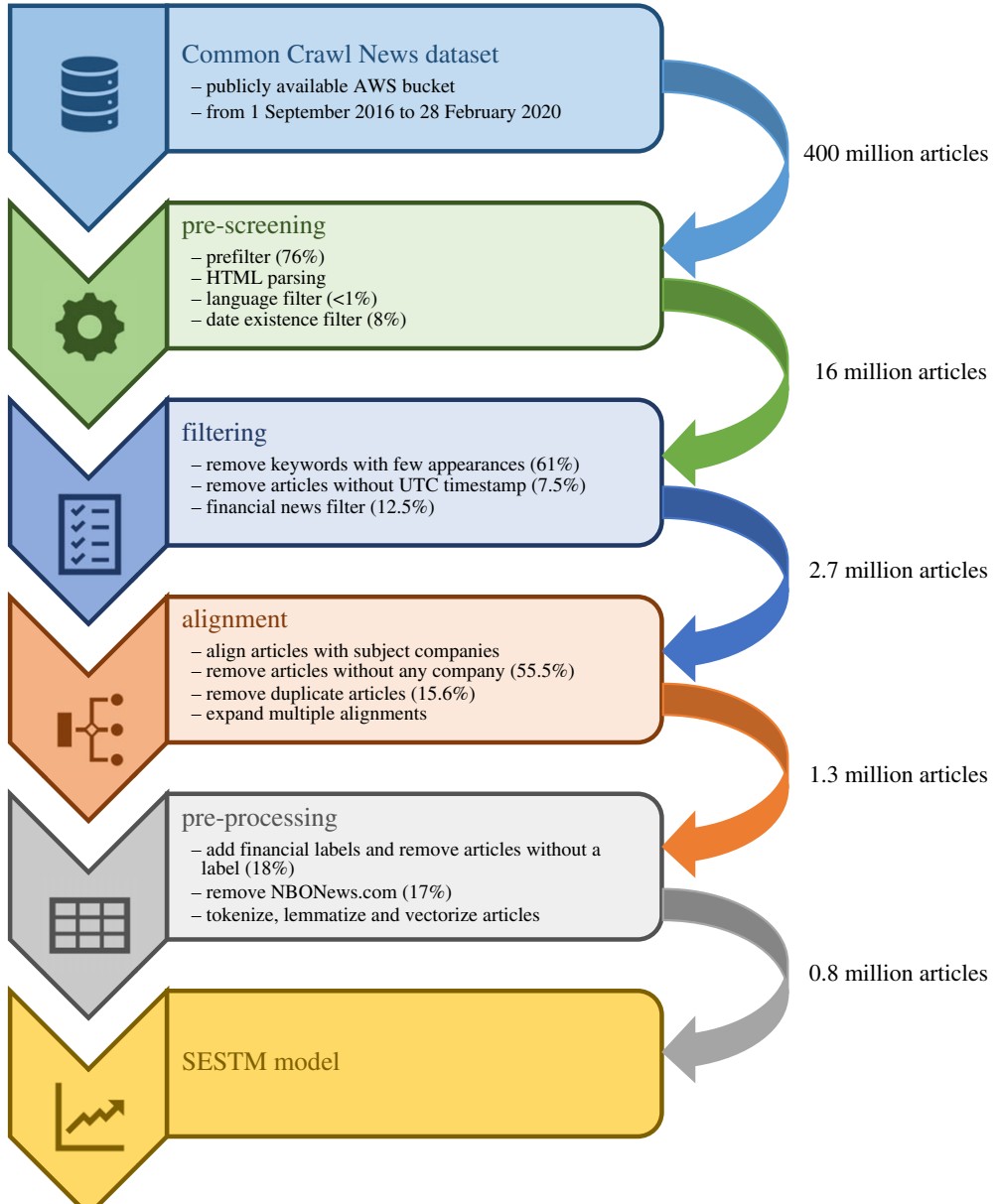

**Figure 1.** The pipeline deployed to process and transform the Common Crawl News dataset into the dataset used by the sentiment model. Each box represents one stage of the pipeline where data transformation and filtering steps are applied. The numbers next to the arrows show how many articles are passed on from one stage to the next. The percentages in the brackets after each filtering step show the proportion of articles removed in that specific step.

processing we had to address several issues including (i) extracting the main text and additional meta-information such as timestamps from the HTML strings of each article, (ii) matching each article to the relevant company names and tickers, (iii) deciding which articles relate to the financial performance of the subject company, and (iv) removing any duplicate news articles present in the data. After the processing, we have a dataset of about 1.3 million financial articles written in English covering S&P 500 companies from August 2016 to February 2020. For each article, we identify its URL, publication time, crawling time, title and all referenced constituent companies in the S&P 500 index. We describe the details of our processing pipeline in the electronic supplementary material, Methods S1 and table S1.

## 2.2. Transfer entropy

To avoid making specific assumptions of the relationship between sentiment and stock returns, rather than using classical Granger causality we use transfer entropy, a model-free measure from information

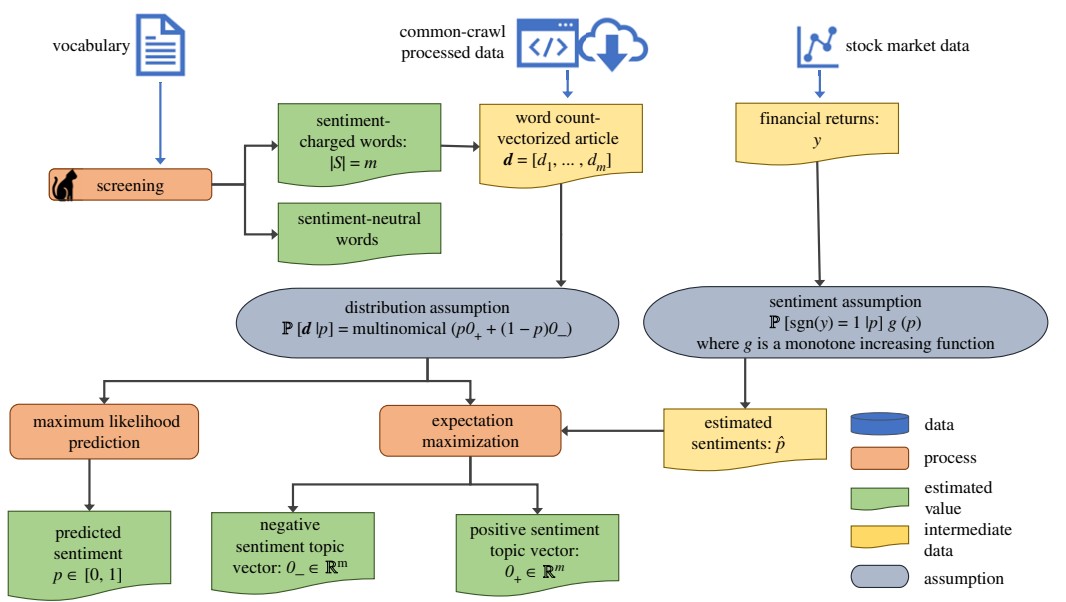

**Figure 2.** Process chart of the sentiment model. The two assumptions underlying the sentiment model are depicted in the middle. The data used in fitting the model is shown at the top. We apply the predicted sentiment scores (bottom-left corner) to analyse transfer entropy and simulate several simple trading strategies.

theory that is not restricted to linear dynamics or Gaussian assumptions [21,56]. For a random variable $X$, the Shannon entropy $H(X) = \mathbb{E}[-\log p(X)]$ measures the expected level of 'information' or 'uncertainty' associated with its outcomes. Intuitively, if the logarithm is expressed in base two then $H(X)$ represents the number of bits of the optimal code length for a lossless data compression of events from data source $X$. In order to quantify information content $H(X_t)$ from a time-dependent stochastic process, $\{X_t\}$, one needs to analyse transition probabilities [57] of the underlying stochastic process. In particular, using the idea of finite-order Markov processes, Schreiber [20] introduced *transfer entropy* (TE), a measure of information transfer of systems evolving through time. For each company in the S&P 500 index, we compute the TE from company news sentiment $\{s_t\}$ to stock returns $\{r_t\}$, defined as

$$TE_{s \to r} := H(r_{t+1}|r_t) - H(r_{t+1}|r_t, s_t), \tag{2.1}$$

where $H(X|Y) := -\sum_{i,j} p(x_i, y_j) \log[p(x_i|y_j)]$ denotes the conditional Shannon entropy [22]. The transfer entropy (2.1) can be expressed as the KL divergence

$$TE_{s \to r} = \sum p(r_{t+1}, r_t^{(m)}, s_t^{(k)}) \log \frac{p(r_{t+1}|r_t^{(m)}, s_t^{(k)})}{p(r_{t+1}|r_t^{(m)})}, \tag{2.2}$$

where we define $s_t^{(k)} := (s_t, \ldots, s_{t-k+1})$ and $r_t^{(m)} := (r_t, \ldots, r_{t-m+1})$, which makes explicit that transfer entropy measures the log deviation from the generalized Markov property $p(r_{t+1}|r_t^{(m)}) = p(r_{t+1}|r_t^{(m)}, s_t^{(k)})$.

## 2.3. News sentiment model

To assign sentiment scores to each news article, in this study we use an approach referred to as sentiment extraction via screening and topic modelling (SESTM) [54]. While closely related to the latent Dirichlet allocation (LDA) [58] and vector-based representations such as word2Vec [59] and GloVe [60]; unlike these models, SESTM is trained in a supervised fashion that facilitates interpretability and provides theoretical guarantees on the accuracy of estimates with minimal assumptions [54]. Figure 2 depicts the main steps of the SESTM model we use in this article. We briefly describe our model below; a more detailed description can be found in the electronic supplementary material, Methods S2.

Before training the SESTM model, we apply standard pre-processing steps from NLP to turn articles into document-term vectors, including stopword-removal, tokenization and lemmatization. For each article $i$ in our dataset, we assign a sentiment score $p_i \in [0, 1]$ that reflects the financial sentiment the article bears towards the subject company, where $p_i = 0$ and $p = 1$ denote the most negative and positive sentiment achievable, respectively. We view a sentiment of $p = 0.5$ as neutral. We assume

positive (negative) sentiment for a company most likely leads to positive (negative) return for the company's stock in the following sense:

$$\mathbb{P}(\text{sgn}(r_i) = 1) = g(p_i), \tag{2.3}$$

where $\mathbb{P}$ denotes the probability, $g(\cdot)$ is a monotonically increasing function and $r_i$ is the corresponding financial return of the company pertinent to news article $i$.

We assume that only a subset of the words in the corpus's dictionary is relevant and refer to these as *sentiment-charged*. The remaining words are referred to as *sentiment-neutral*. The model determines the sentiment-charged vocabulary of words, $S$, by including only those words that occur sufficiently frequently in our corpus and that are predominantly associated with either positive or negative returns. We remove all sentiment-neutral words from our original dictionary.

For each article, we associate a document-term vector, $d_i$, of the occurrences of the sentiment-charged words and assume that it has a mixture multinomial distribution of the form

$$d_i \sim \text{Multinomial}(s_i, p_i O_+ + (1 - p_i)O_-), \tag{2.4}$$

where $s_i = \sum_{j \in S} d_{i,j}$ scales the distribution and $p_i O_+ + (1 - p_i)O_-$ is a mixture of two topics that determines the probability distribution over the sentiment-charged words. $O_+$ describes the probability of the words in a maximally positive article, $p_i = 1$. Similarly, $O_-$ describes the probability of the words in a maximally negative article, $p_i = 0$. We assume that $O_+$, $O_- \in \mathbb{R}_+^{|S|}$ are normalized such that $\|O_+\|_1 = \|O_-\|_1 = 1$. For the articles with sentiment not on the boundary, $0 < p_i < 1$, word frequencies are convex combinations of those from the two topics. We train our model by estimating the vectors $O_+$ and $O_-$ via expectation maximization (EM), denoting their estimates by $\hat{O}_+$ and $\hat{O}_-$.

The sentiment $\hat{p}_i$ associated with the $i$th news article is determined by maximum likelihood estimation (MLE) applied to the multinomial distribution of the news article's document-term vector, $d$. In other words, we determine $\hat{p}$ by solving

$$\hat{p}_i = \arg \max_{p \in [0,1]} \left\{ \frac{1}{s_i} \sum_{j \in S} d_{i,j} \log(p \hat{O}_{j,+} + (1 - p)\hat{O}_{j,-}) + \lambda \log(p(1 - p)) \right\}, \tag{2.5}$$

where $s_i := \sum_{j \in S} d_{i,j}$ and $\lambda > 0$ is a regularization parameter. The choice of regularization is equivalent of imposing a beta prior on the sentiment, thereby pulling the estimated values toward the neutral score ($p = 0.5$).

While our assumptions are the same as the SESTM model [54], we deviate from the original parameter estimation procedure due to the smaller size of our dataset. Specifically, the model parameters are estimated at the beginning of each month based on all the articles observed up until that point.

Then the fitted model is used to predict sentiments during the whole month before being updated again. In addition, we keep the hyperparameters used for the sentiment-charged words selection and the prediction regularization parameter, $\lambda$, fixed rather than considering them as a part of the periodic estimation.

# 3. Results

## 3.1. The Common Crawl and financial news coverage

Using the Common Crawl News, a subset of the Common Crawl exclusively for news articles, we process and extract news articles related to the constituent companies in the S&P 500 index over the time period from 26 August 2016 to 27 February 2020. We choose the end of August 2016 as our starting point because prior periods have insufficient news coverage for the companies in the S&P 500 index. An article is matched to a company if and only if the company is mentioned in the title or the first paragraph (see the electronic supplementary material, Methods S1 for a detailed description of our data processing pipeline).

Figure 3*a* shows the 30 most frequently occurring companies in our dataset as measured by the number of distinct articles that mention each company at least once. Figure 3*b* shows the 30 top news sources as measured by the number of unique articles.[3] Not surprisingly, well-known publicly available financial news websites, such as www.reuters.com, www.seekingalpha.com, www.businessinsider.com and www.cnbc.com appear amongst the most frequent sources. Other frequent

---

[3]We omitted the domain www.nbonews.com. While a frequently occurring source, it is no longer accessible and we were unable to verify its legitimacy. We provide summaries that include nbonews.com in the electronic supplementary material, figure S1.

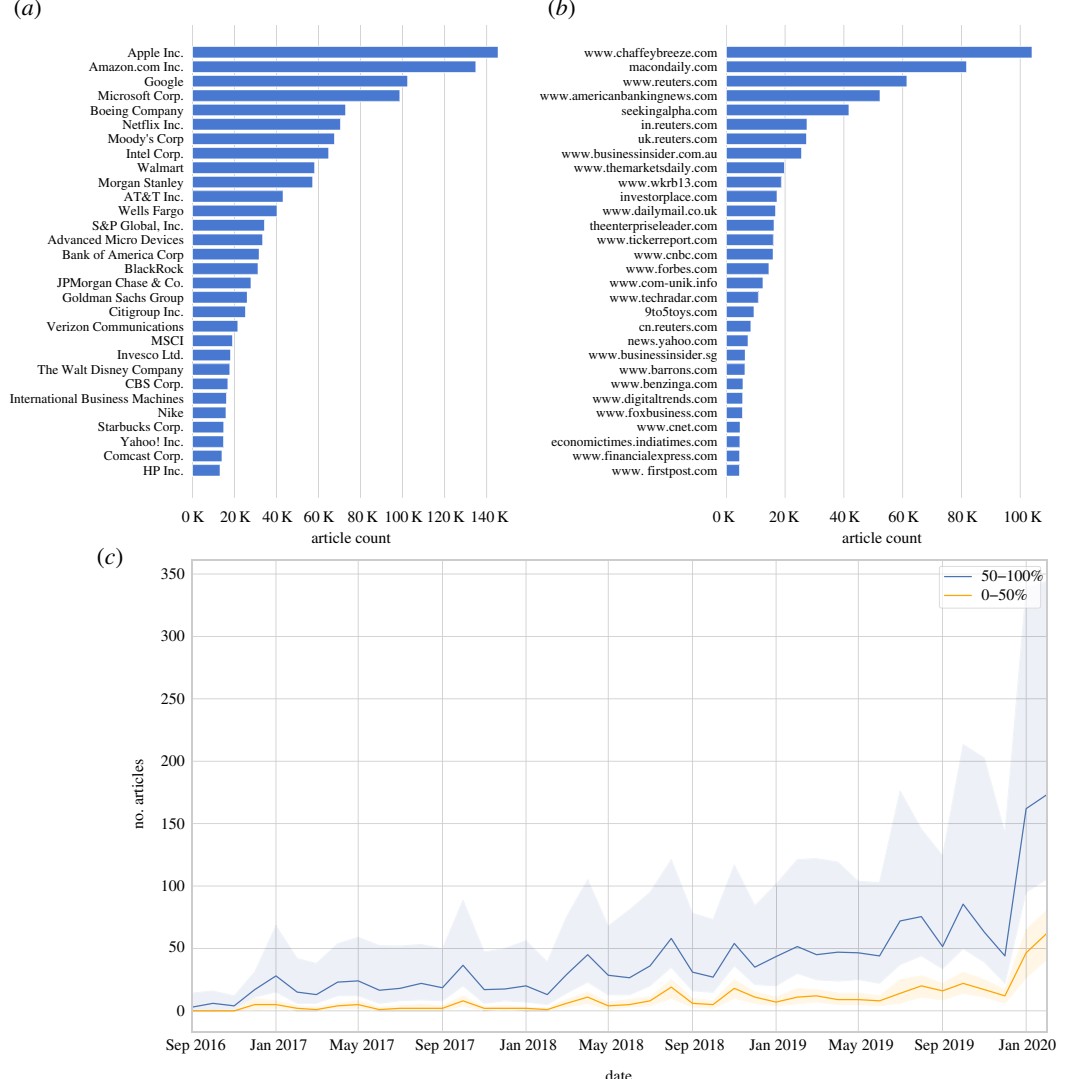

**Figure 3.** Summary of the news dataset used in this article. (*a*) The most frequently mentioned companies as measured by the number of distinct articles. (*b*) The most frequent news sources as measured by the number of distinct articles associated with each source. (*c*) The median number of articles published per company and month. The companies are divided into top and bottom halves by the total number of articles published about them. The shaded regions represent the 25% and 75% percentiles of each half.

sources, including www.chaffeybreeze.com, www.macondaily.com and www.americanbankingnews. com are less well-known. We note that www.chaffeybreeze.com and www.macondaily.com redirect to www.marketbeat.com that publish articles about specific companies and investor ratings. However, as Common Crawl only accesses publicly available sources with free content, any subscription-based news services, such as the Wall Street Journal (https://www.wsj.com) or Barron's (https://www.barrons. com), are not part of our dataset. Figure 3*c* depicts the median number of articles per company published each month throughout our time period. We divide the companies into upper and lower halves by total article count. The shaded regions show the 25% and 75% percentiles for each half by month, respectively. We observe that the distribution of published articles is right skewed as some companies have significant news coverage while others are mentioned less frequently. The shaded percentile regions illustrate that the top 50% of the companies receive significantly more coverage than the bottom 50%. In addition, we emphasize that the amount of news is increasing month over month, as Common Crawl continues to increase the number of websites that are crawled.

## 3.2. Information transfer from news to stock returns

After processing all articles, for each company $i$ in the S&P 500 index we have an inhomogeneous time series of article sentiment scores (from the SESTM model) occurring at irregular timestamps

$\{t_1^i, t_2^i, \ldots, t_n^i\}$, that correspond to the publication times of $n$ articles. For our subsequent analysis, we need the time series of sentiment scores to be occurring at regular time intervals. We achieve this by binning the sentiment series to hourly intervals and taking the average scores inside each bin. For example, for each company $i$ we obtain time series of hourly average sentiment scores $\{s_t^i\}_{t=1}^m$, where $\{1, 2, \ldots, m\}$ represent the hourly binned timestamps of total length $m$. To simplify the notation, we drop the superscript $i$, referencing the company. Hence the hourly price returns of a particular stock will be denoted by $r_t$ (see electronic supplementary material, Data S1, for more details). We characterize the information transfer from the news sentiment series of each company in the S&P 500 index to its stock price returns by computing its transfer entropy as described in formula (2.1). In particular, we use transfer entropy to quantify the amount of uncertainty reduction in the future return, $r_{t+1}$, for each stock given the information of lagged news sentiment and price returns, $(s_t, r_t)$.

To address any non-stationarity of the time series of sentiment scores, we apply the difference operator on our time series and perform an augmented Dickey–Fuller test to detect the presence of unit roots with $p$-values ($<0.01$) obtained through regression surface approximation [61,62].

To compute the statistical significance ($p$-values) of transfer, we employ a non-parametric bootstrap method of the underlying Markov process [56,63] and use effective transfer entropy [64] to perform finite sample corrections.

Using our dataset over the period from 3 January 2018 to 27 February 2020,[4] for each company we compute the transfer entropy between hourly sentiment score differences and hourly price returns. Figure 4a,b displays companies with statistically significant transfer entropy ($p$-value $< 0.01$) and the estimated distribution of $p$-values. We observe that the distribution of $p$-values of the depicted stocks are below the 0.05 level. In the electronic supplementary material, figures S4 and S5, we show the results with the control of false discovery rate (FDR $< 0.05$) and again find the presence of statistically significant reduction of uncertainty for 1 h and 2 h old sentiment signals.

Our analysis corroborates public news contribution to information dissemination and price discovery at different time-scales [65]. The existence of multiple time-scales in financial markets is connected to a heterogeneous market hypothesis [65] and arises from differences in heterogeneity across market participants, including different trading constraints, risk profiles, locations, information processing and decision frequencies [66].

## 3.3. Economical significance of news sentiment from Common Crawl News data

In the previous section, we demonstrated that public news sentiment has a statistically significant impact on the uncertainty reduction of future stock returns. This result poses the question whether this impact is also economically significant. To address this question, we analyse performance of the following simple sentiment-based trading strategy. For each day in our sample, we rank all S&P 500 companies based on their news sentiment scores from news articles published between 9.30 on the previous trading day and 9.00 of the current day, where times reflect Eastern Time (ET) throughout this article. For companies with multiple news articles, we average their news sentiments to obtain a single sentiment score for each company. Daily, we form a portfolio with long positions of equal amounts in the 20 companies with the most positive sentiment scores and short positions of equal amounts in the 20 companies with the most negative sentiment scores.[5] We refer to this daily rebalanced portfolio as the *Day 1 sentiment strategy*, where 'Day 1' denotes the one-day lag of the sentiment scores. We track its daily open-to-open return over time, i.e. from 9.30 on the day of portfolio formation to 9.30 the following day. Similarly, we form Day 0 and Day −1 sentiment portfolios and track their daily returns through time.[6] Note that the Day 0 and Day −1 portfolios are 'look-ahead' portfolios as they rely on receiving news ahead of its publishing time. Nevertheless, we use these for comparison purposes below.

Importantly, we emphasize that our experiment here is not meant to reflect implementable real-world trading strategies used by professional money managers and hedge funds. For that, we would also have to include the analysis of trading costs, something which is beyond the scope of this study. Instead, we

---

[4]A period between August 2016 and December 2017 is used as a 'warm-up' period for the sentiment model. For the exact fitting procedure, see the electronic supplementary material, Methods S2.

[5]A long position refers to the investor having bought and therefore owns shares. A short position refers to the investor having borrowed and sold shares on the open market, anticipating to buy them back later for less money.

[6]Specifically, denoting the current trading day by $t$, we use news from 9.30 on day $t-1$ to 9.00 on day $t$ for the Day 1 portfolio, 9.30 on day $t$ to 9.00 on day $t+1$ for the Day 0 portfolio and 9:30 a.m. on day $t+1$ to 9.00 on day $t+2$ for the Day −1 portfolio. For all portfolios, we enter the market at 9.30 on day $t$ and hold our positions until 9.30 on day $t+1$.

**Figure 4.** (*a*) Companies and corresponding significant Shannon transfer entropy (and effective transfer entropy) from hourly sentiment score differences to hourly price returns. The unit of transfer entropy is bits (logarithm with base 2), corresponding to the reduction of the average optimal code length needed to encode stock returns with lagged sentiment. Transfer entropy was calculated for the period from January 2018 to February 2020 using time series of hourly returns from 9.30 to 15.30 Eastern Time and corresponding lagged average sentiment scores. The statistical significance (*p*-value < 0.01) of transfer entropy was estimated with 300 bootstrap samples and 100 shuffles to obtain the effective transfer entropy. (*b*) Box and whisker plots of estimated distributions of the *p*-values for selected company tickers. The box and whisker plots show Q1, median, Q3, minimum, maximum and estimated outliers.

use a simple trading exercise to study the interplay between public news and financial markets also from an economical point of view.

We compare our sentiment-based trading strategies performance with the SPDR S&P 500 trust (ticker symbol: SPY) and a set of random portfolios as a null-model benchmark. SPY is an exchange-traded fund tracking the S&P 500 index. Each random portfolio is rebalanced daily at the same time as the Day 1 sentiment strategy and consists of long and short legs, each with positions of equal amounts in 20 randomly chosen stocks from the S&P 500 index. We simulate 500 random portfolio histories and use their resulting return series to bootstrap performance metrics.

In figure 5, we depict the cumulative return of the Day 1 sentiment strategy relative to our benchmarks from January 2018 to February 2020. We summarize the performance statistics of the Day 1 sentiment trading strategy and benchmarks in table 1. We use the annualized Sharpe ratio as one of our performance metrics, defined as the annualized average return divided by the annualized volatility of return. Clearly, the Day 1 sentiment strategy outperforms the SPY and random portfolios, obtaining an annualized Sharpe ratio of 1.64 as compared with 0.48 and −0.01 for SPY and random portfolios, respectively. Regressing the returns of the Day 1 sentiment strategy on the returns of SPY, we observe that the intercept, denoted by $\alpha$, and $R^2$ of this regression is 20.69% (annualized) and 0.4%, respectively. $\alpha$ is significant at the 1%-level. We conclude that that the Day 1 sentiment strategy (i) outperforms the market and (ii) is uncorrelated with the market. This supports that there is economically and statistically significant information in public news sources. However, note that contrary to the transfer entropy measure that quantifies the average uncertainty reduction during the

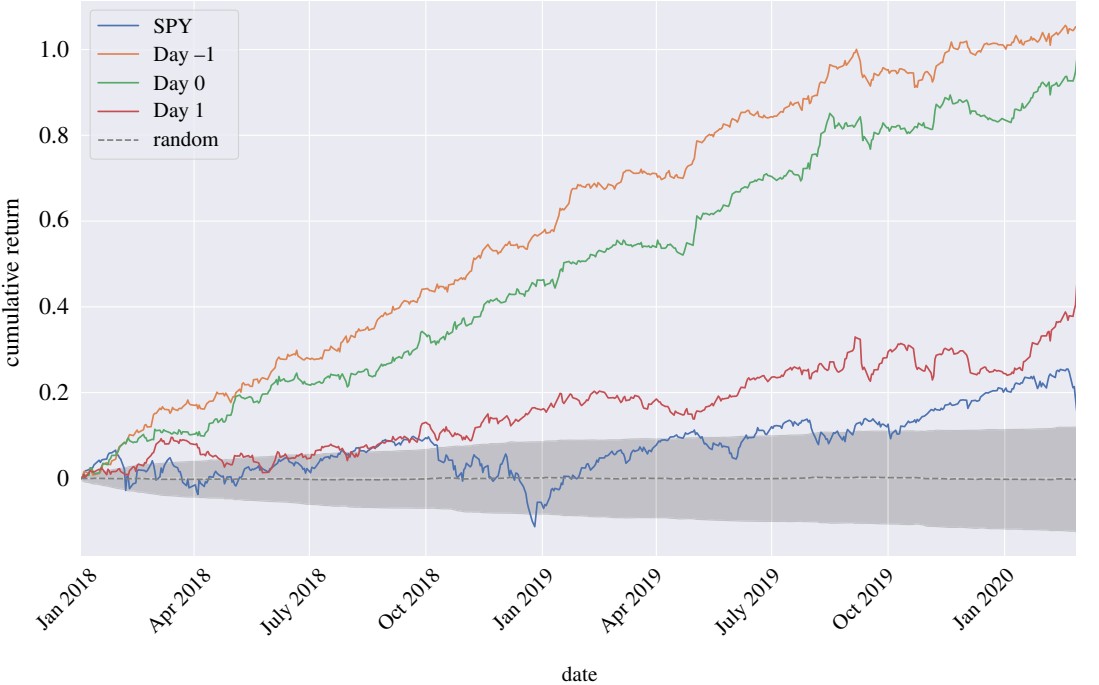

**Figure 5.** Cumulative returns of trading strategies and benchmarks. 'Day 1' represents the cumulative returns of the Day 1 sentiment strategy based on the Common Crawl News dataset from January 2018 to February 2020. SPY is the SPDR S&P 500 trust. 'Random' denotes the average of the random strategies along with 1 s.d. confidence bands obtained from 500 simulations. 'Day 0' and 'Day −1' are the 'look-ahead' sentiment strategies relying on future information.

**Table 1.** Performance statistics of the Day 1 sentiment trading strategy and benchmarks from January 2018 to February 2020. The sentiment trading strategy is based on news articles from the Common Crawl News dataset. SPY is the SPDR S&P 500 trust. 'Random' denotes the baseline strategy where each day we randomly select companies to invest in. 'Day 0' and 'Day −1' are 'look-ahead' sentiment strategies, reported for comparison purposes. Statistics are computed using daily returns ($n = 542$). MDD is the maximum daily drawdown defined as the maximum observed decline from a historical peak of the price until a new peak is attained. The $p$-values $< 0.001$ are denoted with symbol ***, $p$-values $< 0.01$ with symbol **, and $p$-values $< 0.02$ with symbol *. The $p$-values for the Sharpe ratios were bootstrapped from 500 random backtests. We obtain $\alpha$ (the intercept) and $R^2$ by regressing the daily returns of the portfolios on the daily returns of the SPY. The performance metrics of the random portfolios were bootstrapped from 500 random backtests. For daily turnover, we use $(1/2T)\sum_{t=1}^{T}\left(\sum_i |w_{i,t+1} - w_{i,t}(1 + r_{i,t+1})|\right)$, where $w_{i,t}$ denotes portfolio weight at time $t$ and $r_{i,t+1}$ represents daily return at time $t + 1$ of stock $i$.

|  | Day −1 | Day 0 | Day 1 | SPY | random |
|---|---|---|---|---|---|
| annualized average return | 48.95% | 45.42% | 21.02% | 7.25% | −0.11 ± 5.67% |
| annualized volatility | 11.65% | 12.43% | 12.85% | 15.06% | 8.37 ± 0.33% |
| annualized Sharpe ratio | 4.20*** | 3.66*** | 1.64** | 0.48 | −0.01 ± 0.68% |
| MDD | 8.82% | 8.34% | 10.31% | 21.04% | 14.64 ± 6.19% |
| annualized $\alpha$ | 47.88%*** | 45.36%*** | 20.69%* | 0 | 0.05 ± 5.70% |
| $R^2$ | 0.004 | 0.001 | 0.004 | 1 | 0.002 ± 0.004% |
| daily turnover | 82.12% | 82.27% | 82.43% | 0% | 96.32 ± 0.11% |

whole period, the trading strategy is an econometric method that is based on a dynamically rebalanced portfolio that adapts to the arrival of new information through time.

As expected, we note that both the Day −1 and Day 0 strategies outperform the Day 1 strategy, achieving Sharpe ratios of 4.20 and 3.66, respectively. Similarly, they strongly outperform both baselines and display no significant correlation with the market ($R^2$ when regressing on SPY returns

is 0.4% and 0.1%, respectively). While these two 'look-ahead' strategies rely on future information, they provide additional support to that there is significant correlation between stock returns and sentiment derived from the Common Crawl News dataset.

## 3.4. Comparing the information transfer of private and public news to financial markets

We investigate to what extent there is a difference in the information transfer of publicly available news as compared with commercially available news data to financial security prices. As noted above, the Common Crawl News data consists of only freely available public news. That is, the dataset does not contain news content from web pages and news providers that are subscription-based or require registration.

To measure the effect of non-public news sources, we obtained the Alexandria dataset,[7] a commercial news database consisting of financial news curated from about a dozen subscription-based data sources, including Dow Jones News Wire, Wall Street Journal and Barrons. Alexandria uses a proprietary algorithm to associate each news article to the companies in question and to assign sentiment scores.

We use the same simple trading strategy as above to build daily sentiment portfolio using the Alexandria dataset and compare its trading results with those based on the Common Crawl News dataset. The trading strategy using the Alexandria dataset obtains a Sharpe ratio of 1.51 over our time horizon (other performance metrics are available in the electronic supplementary material, table S3). Importantly, the correlation of the return series of the two strategies is only 0.07 ($p$-value $< 0.1$). That the correlation is not statistically different from zero suggests the sentiment scores derived from the Common Crawl News and Alexandria datasets are based on different underlying information. In fact, the average Jaccard index,[8] between the long and short stock positions based on the sentiment scores from the Alexandria and Common Crawl News datasets are $0.020 \pm 0.022$ and $0.019 \pm 0.023$, respectively. This means that the overlap of companies in the portfolios formed on the Alexandria and Common Crawl News sentiment scores is less than one on average.

We conclude there is valuable information present in both datasets to predict future returns. However, the information from the datasets are different (see electronic supplementary material, figure S3) and the Common Crawl News dataset provides complementary information to that of the Alexandria dataset (see electronic supplementary material, Discussion S1). There are two main reasons for why the datasets are different: (i) they use different news sources and (ii) the computed sentiment scores are determined from different models. Alexandria relies on subscription-based financial news, whereas Common Crawl only accesses publicly available sources. Alexandria deploys a proprietary ML approach to compute sentiment scores for each news article, whereas we use the SESTM model to determine sentiment scores for the news articles from the Common Crawl News dataset.

## 4. Discussion

Processing roughly 400 million articles from the Common Crawl News data comes with many non-trivial engineering challenges, including parsing different HTML formats used on the websites, identifying and removing duplicate articles, aligning each article and financial return to corresponding company. The SESTM sentiment model was chosen as a consensus of the complexity, interpretability and theoretical foundations in supervised learning and topic modelling in NLP (see electronic supplementary material, Discussion S3 for details). In the electronic supplementary material, table S2 and figure S2, we explore the effects of deep learning models for sentiment extraction, using the pre-trained [67] bidirectional encoder representations from transformers model (BERT) [68]. It is important to emphasize that the focus of this paper is not on what the best sentiment model is, but rather on the analysis of the interaction of news from the World Wide Web and the financial market as a prototype of an efficient 'information absorbing' system. By analysing the time-series of sentiment scores and price returns, we find evidence of statistically significant transfer of information on the intraday level over the period from January 2018 to February 2020. In this study, we did not analyse possible confounding effects [69,70] between multivariate time series of sentiment and stock price returns, as

---

[7]See https://www.alexability.com.

[8]The Jaccard index, a measure of the similarity between two discrete sets $A$ and $B$, is defined as $J(A, B) := |A \bigcap B|/|A \bigcup B|$. It takes values in the range $[0, 1]$. The higher the index, the greater the similarity between the two sets.

we were not focused on causal inference [71,72] but on bivariate transfer of information between news to corresponding stocks. The multivariate case of transfer entropy can be further extended with partial transfer entropy [69,73] in future work. By using a simple sentiment-based trading strategy as an econometric tool only, and not with the intention of being a realistic and implementable trading strategy, we find that our sentiment signals from public news are carrying both economic value and complementary information compared with non-public and commercial news data providers. Our analysis provides support for that public news contributes to information dissemination and price discovery at different time-scales during a single trading day.

Data accessibility. Data and relevant code for this research work are stored in GitHub: https://github.com/ffaltings/news_and_markets/tree/v0.1 and have been archived within the Zenodo repository: https://doi.org/10.5281/zenodo.5013220.

Authors' contributions. F.F., M.J. and B.P. contributed equally to the work. All authors contributed to the writing and editing of the manuscript. P.N.K. and N.A.-F. designed and supervised this project. F.F., M.J. and B.P. processed the Common Crawl News data. All authors contributed to the modelling and analysis.

Competing interests. We declare we have no competing interests.

Funding. N.A.-F. acknowledges financial support from SoBigData++ through Grant Agreement no. 871042. M.J. acknowledges financial support from Public Scholarship and Development Fund of the Republic of Slovenia.

Acknowledgements. F.F., M.J. and B.P thank Prof. Zhang Ce and Prof. Andreas Krause for their helpful comments and AWS credits during the Data Science Lab course.

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
