## [Peer Review File · Royal Society Open Science]

Review History

RSOS-202321.R0 (Original submission)

Review form: Reviewer 1

Is the manuscript scientifically sound in its present form?

Yes

Are the interpretations and conclusions justified by the results?

Yes

Is the language acceptable?

Yes

Do you have any ethical concerns with this paper?

No

Have you any concerns about statistical analyses in this paper?

No

Recommendation?

Major revision is needed (please make suggestions in comments)

Comments to the Author(s)

Overall, the manuscript is well-written. However, the message to me seems a bit too narrow. Common Crawl and the processing of its news articles seems interesting using your approaches. I would urge the authors to think of what they want to achieve with this paper. I think to be in RSOS, the paper must also compare with likewise contributions in NLP conferences, albeit with less technical NLP jargon.

Q1. The claim that the finBERT model was weaker than your Common Crawl News model seems unconvincing. Please elaborate on that.

Q2. The following papers were not cited or mentioned. How do they compare with your work? These papers are closer to the machine learning and NLP community, I suggest you provide a reason why their work is not relevant.

Ding, Xiao, et al. "Deep learning for event-driven stock prediction." Twenty-fourth international joint conference on artificial intelligence. 2015.

Luss, Ronny, and Alexandre d'Aspremont. "Predicting abnormal returns from news using text classification." *Quantitative Finance* 15.6 (2015): 999-1012.

Q3. The Ding et al. paper seems to be aimed at a more finance journal type of publication. Your paper was submitted to RSOS. Do you have some idea or thought on submitting to financial or NLP venues as alternatives?

Q4. I couldn't find the average change in the portfolio compositions. Do the long or short positions change daily? Please elaborate on this or provide a statistic on how portfolio compositions change daily.

Q5. For the paper to be competitive, you might want to compare to the two papers above and strongly state why deep learning was not used. To me it seems, with such a large data set, deep learning sentiment models would work better. Computational cost is no excuse to downplay finBERT. Perhaps a new algorithm slightly tweaked might work on your model.

Q6. Not using paid subscriptions in your datasets seems to overlook a key part of the financial news media. If a hedge fund manager wanted to implement your strategy, they would surely pay for WSJ, FT and Bloomberg. These are essential tools for a fund manager. Your claim that using only publicly available information from previous day is better or sufficient is hard to believe.

Q7. A bit more detail on the profit generated. It is my understanding that a portfolio is rebalanced the next day with positions closed out. Were opening prices used? I assume the positions are small so taking positions does not cause adverse moves? In reality there are always slippage costs. How would you account for this.

Q8. What about bid-ask spreads? Shouldn't such a cost be added to your position? For the Sharpe ratio, was regular calendar time or business calendar time used?

The last two questions are more from a finance perspective and so probably not as essential as the need to address the machine learning or NLP queries.

Review form: Reviewer 2

Is the manuscript scientifically sound in its present form?

Yes

Are the interpretations and conclusions justified by the results?

Yes

Is the language acceptable?

Yes

Do you have any ethical concerns with this paper?

No

Have you any concerns about statistical analyses in this paper?

No

Recommendation?

Accept with minor revision (please list in comments)

Comments to the Author(s)

This article takes a new stab at an old problem: How does news affect stock prices.

It uses data from Common Crawl to extract finance-related news articles and extract news sentiments for stocks in S&P500.

They then use transfer entropy to estimate how much the returns followed the news sentiment at the intra-day scale.

finally, they pick the stocks most affected by the news and use the top 20 positive and negative ones to form a simple trading strategy. They show that this trading strategy outperforms both the ETF SPY and random strategies by a large margin.

I enjoyed reading this paper. While a great deal of work has been done on correlation of news sentiment with stock prices and using transfer entropy in stock markets, this way of combining the two and eventually verifying the predictions using a trading strategy seems novel to me. If the authors can address my concerns, I recommend the paper for publication.

My main concern is the very high profit margin of your strategy.

As you may understand, profit of 21% is mind-blowingly high for day trading strategies. While I don't think it's impossible, usually even 2 sigma above random is quite an achievement. That's why I urge you to think about confounding factors. Could it be after-hour movements, encoded in news sentiments? Is it driven by the wild steady growth of the stock market from 2018-2020? What else can be confounding the result?

Below, I have a few minor questions and comments.

Eq 2.2: what is the sum over? t , m and k ?

p6: MLE for \hat{p}_i : how difficult is the EM?

p7 line 54: why average sentiment, instead of sum? For each article you were summing word sentiments, so when binning, wouldn't it make more sense to sum within bins?

p8: describing Day 1: there can be after-hour movements reported in the news, but showing up in the next day's prices. Can that be a significant contribution to the success of your strategy? What other confounding factors can you think of?

Decision letter (RSOS-202321.R0)

Dear Dr Antulov-Fantulin

The Editors assigned to your paper RSOS-202321 "On the impact of publicly available news and information transfer to financial markets" have now received comments from reviewers and would like you to revise the paper in accordance with the reviewer comments and any comments from the Editors. Please note this decision does not guarantee eventual acceptance.

Please submit your revised manuscript and required files (see below) no later than 21 days from today's (ie 09-Mar-2021) date. Note: the ScholarOne system will 'lock' if submission of the revision is attempted 21 or more days after the deadline. If you do not think you will be able to meet this deadline please contact the editorial office immediately.

on behalf of Professor Marta Kwiatkowska (Subject Editor)
openscience@royalsociety.org

Associate Editor Comments to Author:

A number of comments and queries have been raised about, for instance, the relative similarity to an earlier work (at least in using a similar model). It would be instructive if the authors could address this - as well as the other comments from the reviewers - in their revision.

Reviewer comments to Author:

Reviewer: 1

Comments to the Author(s)

Overall, the manuscript is well-written. However, the message to me seems a bit too narrow. Common Crawl and the processing of its news articles seems interesting using your approaches. I would urge the authors to think of what they want to achieve with this paper. I think to be in RSOS, the paper must also compare with likewise contributions in NLP conferences, albeit with less technical NLP jargon.

Q1. The claim that the finBERT model was weaker than your Common Crawl News model seems unconvincing. Please elaborate on that.

Q2. The following papers were not cited or mentioned. How do they compare with your work? These papers are closer to the machine learning and NLP community, I suggest you provide a reason why their work is not relevant.

Ding, Xiao, et al. "Deep learning for event-driven stock prediction." Twenty-fourth international joint conference on artificial intelligence. 2015.

Luss, Ronny, and Alexandre d'Aspremont. "Predicting abnormal returns from news using text classification." *Quantitative Finance* 15.6 (2015): 999-1012.

Q3. The Ding et al. paper seems to be aimed at a more finance journal type of publication. Your paper was submitted to RSOS. Do you have some idea or thought on submitting to financial or NLP venues as alternatives?

Q4. I couldn't find the average change in the portfolio compositions. Do the long or short positions change daily? Please elaborate on this or provide a statistic on how portfolio compositions change daily.

Q5. For the paper to be competitive, you might want to compare to the two papers above and strongly state why deep learning was not used. To me it seems, with such a large data set, deep learning sentiment models would work better. Computational cost is no excuse to downplay finBERT. Perhaps a new algorithm slightly tweaked might work on your model.

Q6. Not using paid subscriptions in your datasets seems to overlook a key part of the financial news media. If a hedge fund manager wanted to implement your strategy, they would surely pay for WSJ, FT and Bloomberg. These are essential tools for a fund manager. Your claim that using only publicly available information from previous day is better or sufficient is hard to believe.

Q7. A bit more detail on the profit generated. It is my understanding that a portfolio is rebalanced the next day with positions closed out. Were opening prices used? I assume the positions are small so taking positions does not cause adverse moves? In reality there are always slippage costs. How would you account for this.

Q8. What about bid-ask spreads? Shouldn't such a cost be added to your position? For the Sharpe ratio, was regular calendar time or business calendar time used?

The last two questions are more from a finance perspective and so probably not as essential as the need to address the machine learning or NLP queries.

Reviewer: 2

Comments to the Author(s)

This article takes a new stab at an old problem: How does news affect stock prices. It uses data from Common Crawl to extract finance-related news articles and extract news sentiments for stocks in S&P500.

They then use transfer entropy to estimate how much the returns followed the news sentiment at the intra-day scale.

finally, they pick the stocks most affected by the news and use the top 20 positive and negative ones to form a simple trading strategy. They show that this trading strategy outperforms both the ETF SPY and random strategies by a large margin.

I enjoyed reading this paper. While a great deal of work has been done on correlation of news sentiment with stock prices and using transfer entropy in stock markets, this way of combining the two and eventually verifying the predictions using a trading strategy seems novel to me.

If the authors can address my concerns, I recommend the paper for publication.

My main concern is the very high profit margin of your strategy.

As you may understand, profit of 21% is mind-blowingly high for day trading strategies. While I don't think it's impossible, usually even 2 sigma above random is quite an achievement. That's why I urge you to think about confounding factors. Could it be after-hour movements, encoded in news sentiments? Is it driven by the wild steady growth of the stock market from 2018-2020? What else can be confounding the result?

Below, I have a few minor questions and comments.

Eq 2.2: what is the sum over? t , m and k ?

p6: MLE for \hat{p}_i : how difficult is the EM?

p7 line 54: why average sentiment, instead of sum? For each article you were summing word sentiments, so when binning, wouldn't it make more sense to sum within bins?

p8: describing Day 1: there can be after-hour movements reported in the news, but showing up in the next day's prices. Can that be a significant contribution to the success of your strategy? What other confounding factors can you think of?

===PREPARING YOUR MANUSCRIPT===

While not essential, it will speed up the preparation of your manuscript proof if accepted if you format your references/bibliography in Vancouver style (please see

<https://royalsociety.org/journals/authors/author-guidelines/#formatting>). You should include DOIs for as many of the references as possible.

===PREPARING YOUR REVISION IN SCHOLARONE===

Author's Response to Decision Letter for (RSOS-202321.R0)

See Appendix A.

RSOS-202321.R1 (Revision)

Review form: Reviewer 1

Is the manuscript scientifically sound in its present form?

Yes

Are the interpretations and conclusions justified by the results?

Yes

Is the language acceptable?

Yes

Do you have any ethical concerns with this paper?

No

Have you any concerns about statistical analyses in this paper?

No

Recommendation?

Accept as is

Comments to the Author(s)

Thank you. I urge the authors to emphasize their paper is not advocating a profitable trading strategy but rather the interplay between public information and the markets. This link is tenuous and can be better explained or motivated.

Review form: Reviewer 2

Is the manuscript scientifically sound in its present form?

Yes

Are the interpretations and conclusions justified by the results?

Yes

Is the language acceptable?

Yes

Do you have any ethical concerns with this paper?

No

Have you any concerns about statistical analyses in this paper?

No

Recommendation?

Accept as is

Comments to the Author(s)

I am satisfied with the response and recommend the paper for publication.

One clarification regarding my question on summing sentiments: I meant summing per article sentiments. Each article's sentiment of course needs to be normalized by the word count. But the mean sentiment makes sense.

I think the look-ahead effect is also excluded properly in your method and the clarification added in the paper regarding bivariate correlation and not causal effects seems adequate.

Decision letter (RSOS-202321.R1)

Dear Dr Antulov-Fantulin

On behalf of the Editors, we are pleased to inform you that your Manuscript RSOS-202321.R1 "On the impact of publicly available news and information transfer to financial markets" has been accepted for publication in Royal Society Open Science subject to minor revision in accordance with the referees' reports. Please find the referees' comments along with any feedback from the Editors below my signature.

Please submit your revised manuscript and required files (see below) no later than 7 days from today's (ie 09-Jun-2021) date. Note: the ScholarOne system will 'lock' if submission of the revision

is attempted 7 or more days after the deadline. If you do not think you will be able to meet this deadline please contact the editorial office immediately.

on behalf of Marta Kwiatkowska (Subject Editor)
openscience@royalsociety.org

Associate Editor Comments to Author:

Comments to the Author:

Please make sure you incorporate the final suggestions of the reviewers in a final revision.

Acceptance will be contingent on this.

Reviewer comments to Author:

Reviewer: 1

Comments to the Author(s)

Thank you. I urge the authors to emphasize their paper is not advocating a profitable trading strategy but rather the interplay between public information and the markets. This link is tenuous and can be better explained or motivated.

Reviewer: 2

Comments to the Author(s)

I am satisfied with the response and recommend the paper for publication.

One clarification regarding my question on summing sentiments: I meant summing per article sentiments. Each article's sentiment of course needs to be normalized by the word count. But the mean sentiment makes sense.

I think the look-ahead effect is also excluded properly in your method and the clarification added in the paper regarding bivariate correlation and not causal effects seems adequate.

===PREPARING YOUR MANUSCRIPT===

===PREPARING YOUR REVISION IN SCHOLARONE===

- If you are providing image files for potential cover images, please upload these at this step, and inform the editorial office you have done so. You must hold the copyright to any image provided.
- A copy of your point-by-point response to referees and Editors. This will expedite the preparation of your proof.

- Ensure that your data access statement meets the requirements at <https://royalsociety.org/journals/authors/author-guidelines/#data>. You should ensure that you cite the dataset in your reference list. If you have deposited data etc in the Dryad repository, please only include the 'For publication' link at this stage. You should remove the 'For review' link.
- If you are requesting an article processing charge waiver, you must select the relevant waiver option (if requesting a discretionary waiver, the form should have been uploaded at Step 3 'File upload' above).
- If you have uploaded ESM files, please ensure you follow the guidance at <https://royalsociety.org/journals/authors/author-guidelines/#supplementary-material> to include a suitable title and informative caption. An example of appropriate titling and captioning may be found at https://figshare.com/articles/Table_S2_from_Is_there_a_trade-off_between_peak_performance_and_performance_breadth_across_temperatures_for_aerobic_scooping_in_teleost_fishes_/3843624.

Author's Response to Decision Letter for (RSOS-202321.R1)

See Appendix B.

Decision letter (RSOS-202321.R2)

Dear Dr Antulov-Fantulin,

I am pleased to inform you that your manuscript entitled "On the impact of publicly available news and information transfer to financial markets" is now accepted for publication in Royal Society Open Science.

on behalf of Marta Kwiatkowska (Subject Editor)
openscience@royalsociety.org

Appendix A

We have prepared a new revised version taking into account the referee's comments. In the following, we reply point by point to all comments raised by the reviewers.

Reviewer: 1

Comments to the Author(s)

Overall, the manuscript is well-written. However, the message to me seems a bit too narrow. Common Crawl and the processing of its news articles seems interesting using your approaches. I would urge the authors to think of what they want to achieve with this paper. I think to be in RSOS, the paper must also compare with likewise contributions in NLP conferences, albeit with less technical NLP jargon.

Answer: We thank the reviewer for raising the question about our contributions and relation to Natural Language Processing (NLP).

The primary aim of our manuscript is to study **information dynamics** in complex financial systems, that fall to the intersection of the following subjects: complexity, artificial intelligence, applied math & mathematical finance, defined by **Royal Society Open Science journal** at <https://royalsocietypublishing.org/rsos/collection>. Furthermore, in our manuscript we are not claiming to propose another NLP-based model for extracting sentiment signals.

We are well aware that **the Royal Society Open Science** is a mutli-disciplinary journal and hence not specifically focused on the NLP-domain. As our manuscript is not on NLP per se, but rather on information dynamics, we think it is well-suited for RSOS. While we are citing NLP-based models in general, and the specific ones we are using in our work, we use them as tools to understand information dynamics in complex financial systems.

Additionally, we emphasize that in the supplementary information (see Supp. Disc. S3), we include a detailed discussion of related work, including that of NLP-based research on sentiment as well as references suggested by the RSOS reviewers.

Q1. The claim that the finBERT model was weaker than your Common Crawl News model seems unconvincing. Please elaborate on that.

Answer: It was not our intention to claim that finBERT is weaker than the SESTM model. We have edited our manuscript to address this confusion. Please also see our answer to **Q5** below, where we analyze and compare finBERT and SESTM models in more detail.

Q2. The following papers were not cited or mentioned. How do they compare with your work? These papers are closer to the machine learning and NLP community, I suggest you provide a reason why their work is not relevant.

Ding, Xiao, et al. "Deep learning for event-driven stock prediction." Twenty-fourth international joint conference on artificial intelligence. 2015.

Luss, Ronny, and Alexandre d'Aspremont. "Predicting abnormal returns from news using text classification." Quantitative Finance 15.6 (2015): 999-1012.

Answer: We would like to thank the reviewer for these useful references. Ding et al. (2015) use deep learning to learn event embeddings from titles in news articles from Reuters and Bloomberg. Then they apply a CNN to forecast stock performance with news articles. Luss et al. (2015) use SVMs over BOW text features (from Newswire press release) and stock return histories to predict stock price movement (up or down) and abnormality (movement outside the 75th quantile). We now cite them in the related work section of our manuscript.

Needless to say, our work is different in two important ways. First, we are interested in how the information in publicly available news impacts financial markets. Second, we use two more recent models as baselines in our work: the SESTM and finBERT models (Kelly et al. 2019; Devlin et al. 2018).

Q3. The Ding et al. paper seems to be aimed at a more finance journal type of publication. Your paper was submitted to RSOS. Do you have some idea or thought on submitting to financial or NLP venues as alternatives?

Answer: As we mentioned above, the primary aim of this manuscript is the study of information dynamics on financial markets, from a complex systems perspective where machine learning models are used as tools. Therefore, we believe that the RSOS journal is more suitable for our manuscript than a financial or NLP-focused journal.

Q4. I couldn't find the average change in the portfolio compositions. Do the long or short positions change daily? Please elaborate on this or provide a statistic on how portfolio compositions change daily.

Answer: Yes, our portfolio positions are changed on a daily basis. For more details about our portfolio construction methodology, please refer to the section "Economical significance of news sentiment from Common Crawl News data." Additionally, in the updated version of our manuscript we have included statistics (average daily turnover) to relevant tables.

Q5. For the paper to be competitive, you might want to compare to the two papers above and strongly state why deep learning was not used. To me it seems, with such a large data set, deep learning sentiment models would work better. Computational cost is no excuse to downplay finBERT. Perhaps a new algorithm slightly tweaked might work on your model.

Answer: The reviewer suggested two interesting papers (Ding et al. 2015, Luss et al. 2015) that are in the area of NLP and finance. We agree that the suggested references are quite

relevant, and thus we have included them as references in our manuscript. We also clarified above how our work is different from these papers.

Based on the reviewer’s suggestion, we performed an analysis comparing the finBERT model (Devlin et al. 2018, Araci et al. 2019) to SESTM on our data. finBERT is a transformer-based deep learning NLP model. We have included this analysis in the supplementary information (Supplementary Table S2), but reproduce it here:

	Common Crawl (finBERT)				
	Day -1	Day 0	Day 1	SPY	Random
Ann. avg. return	108.51%	134.37%	16.68%	7.25%	$-0.11 \pm 5.67\%$
Ann. volatility	12.77%	13.40%	12.25%	15.06%	$8.37 \pm 0.33\%$
Ann. Sharpe ratio	8.50***	10.03***	1.36*	0.48	-0.01 ± 0.68
MDD	4.93%	7.96%	13.17%	21.04%	$14.64 \pm 6.19\%$
Ann. α	108.41%***	134.30%***	16.79%*	0	$0.05 \pm 5.70\%$
R^2	0.015	0.027	0.002	1	0.002 ± 0.004
Daily turnover	81.44%	81.60 %	81.78 %	0 %	$96.32 \pm 0.11\%$

Supplementary Table S2. Performance statistics of the Day 1 sentiment trading strategy and benchmarks from January 2018 through February 2020. The sentiment trading strategy is based on news articles from the Common Crawl News dataset and sentiment is extracted using finBERT. SPY is the SPDR S&P 500 trust. “Random” denotes the baseline strategy where each day we randomly select companies to invest in. “Day 0” and “Day -1” are “look-ahead” sentiment strategies, reported for comparison purposes. Statistics are computed using daily returns (n=251). MDD is the Maximum Daily Drawdown defined as the maximum observed decline from a historical peak of the price until a new peak is attained. The p-values < 0.001 are denoted with symbol ***, p-values < 0.01 with symbol **, and p-values < 0.05 with symbol *. The p-values for the Sharpe ratios were bootstrapped from 500 random backtests. We obtain α (the intercept) and R^2 by regressing the daily returns of the portfolios on the daily returns of the SPY. Similarly, the performance metrics of the random portfolios were bootstrapped from 500 random backtests.

	Day -1	Day 0	Day 1	SPY	Random
Ann. avg. return	48.95%	45.42%	21.02%	7.25%	$-0.11 \pm 5.67\%$
Ann. volatility	11.65%	12.43%	12.85%	15.06%	$8.37 \pm 0.33\%$
Ann. Sharpe ratio	4.20***	3.66***	1.64**	0.48	-0.01 ± 0.68
MDD	8.82%	8.34%	10.31%	21.04%	$14.64 \pm 6.19\%$
Ann. α	47.88%***	45.36%***	20.69%*	0	$0.05 \pm 5.70\%$
R^2	0.004	0.001	0.004	1	0.002 ± 0.004
Daily turnover	82.12%	82.27 %	82.43 %	0 %	$96.32 \pm 0.11\%$

Table 1. Performance statistics of the Day 1 sentiment trading strategy and benchmarks from January 2018 through February 2020. The sentiment trading strategy is based on news articles from the Common Crawl News dataset. SPY is the SPDR S&P 500 trust. “Random” denotes the baseline strategy where each day we randomly select companies to invest in. “Day 0” and “Day -1” are “look-ahead” sentiment strategies, reported for comparison purposes. Statistics are computed using daily returns (n=542). MDD is the Maximum Daily Drawdown defined as the maximum observed decline from a historical peak of the price until a new peak is attained. The p-values < 0.001 are denoted with symbol ***, p-values < 0.01 with symbol **, and p-values < 0.02 with symbol *. The p-values for the Sharpe ratios were bootstrapped from 500 random backtests. We obtain α (the intercept) and R^2 by regressing the daily returns of the portfolios on the daily returns of the SPY. The performance metrics of the random portfolios were bootstrapped from 500 random backtests. For daily turnover, we use $\frac{1}{2T} \sum_{t=1}^T (\sum_i |w_{i,t+1} - w_{i,t}(1 + r_{i,t+1})|)$, where $w_{i,t}$ denotes portfolio weight at time t and $r_{i,t+1}$ represents daily return at time $t + 1$ of stock i .

Comparing these results to those for the SESTM model (Table 1 in our manuscript; reproduced here for convenience), we highlight the following observations:

- Results in Table 1 and Supplementary Table S2 are not designed to test directly the accuracy of sentiment predictions for different NLP models. Rather the results illustrate the hypothetical performance of simple trading strategies that are based on different information content and time-lags.
- Notably, the finBERT model obtains a better annualized Sharpe ratio of 10.03 for the Day 0 strategy; versus an annualized Sharpe ratio of 3.66 for the SESTM model. This result suggests that as far as “pure” NLP-based performance, finBERT does a better job than SESTM.
- The SESTM model obtains a better annualized Sharpe ratio of 1.64 for Day 1 strategy; versus an annualized Sharpe ratio of 1.36 for the finBERT model. Importantly, the Day 1 strategy relies on the *one-day lag* of the sentiment scores. E.g. for the current trading day t , we use news sentiment from 9:30 a.m. on day $t-1$ through 9:00 a.m. on day t for the Day 1 portfolio. However, we emphasize that better *sentiment extraction* at day $t-1$ does not directly imply better portfolio creation at day t . For instance, the sentiment extracted by finBERT could be more short-lived than that of SESTM.
- Finally, in the Supplementary Figure S3 we provide the correlations of returns of the same simple trading strategy, constructed from different models. In particular, we observe that the correlation between the strategies based on the daily finBERT and SESTM models is large (>0.35), suggesting that the different models are extracting similar (but not the same) information content.

	Common Crawl (SESTM)			Alexandria		
	Weekly	Daily 9.30AM	Daily 4.00PM	Weekly	Daily 9.30AM	Daily 4.00PM
Ann. avg. return	23.30%	21.02%	13.94%	-6.15%	14.87%	23.24%
Ann. volatility	13.77%	12.97%	12.85%	8.16%	9.84%	9.47%
Ann. Sharpe ratio	1.69	1.64	1.08	-0.755	1.51	2.46
MDD	12.41%	10.31%	11.10%	20.16%	7.27%	7.87%
Ann. α	23.26%	20.69%	13.38%	-5.83%	15.22%	23.58%
R^2	0.0002	0.0038	0.0093	0.0078	0.005	0.0093

Supplementary Table S3. Trading performance statistics for different rebalancing rules based on sentiment scores from the Alexandria and Common Crawl News datasets from January 2018 through February 2020. Denoting current trading day by day_t , time intervals from which we consider news for portfolio construction are 9:30 a.m. on day $t - 5$ through 9:00 a.m. on day t for *Weekly* rebalancing, 9:30 a.m. on day $t - 1$ through 9:00 a.m. on day t for *Daily-9.30AM* rebalancing and 4:00 p.m. on day $t - 1$ through 9:00 a.m. on day t for *Daily-4.00PM* rebalancing. In all simulations, we enter positions at 9.30 a.m. on day t . For *Weekly* and *Daily* rebalancing we hold positions until 9.30 a.m. on day $t + 5$ and 9.30 a.m. on day $t + 1$, respectively. Statistics are computed using daily returns ($n=542$). MDD is the Maximum Daily Drawdown defined as the maximum observed decline from a historical peak of the price until a new peak is attained. We obtain α (the intercept) and R^2 by regressing the daily returns of the portfolios on the daily returns of the SPY.

Q6. Not using paid subscriptions in your datasets seems to overlook a key part of the financial news media. If a hedge fund manager wanted to implement your strategy, they would surely pay for WSJ, FT and Bloomberg. These are essential tools for a fund

manager. Your claim that using only publicly available information from the previous day is better or sufficient is hard to believe.

Answer: The Common Crawl only crawls publicly available websites. Importantly, we make no claim that public news is better than paid sources. In section “Comparing the information transfer of private and public news to financial markets” we demonstrate that sentiment signals based on (1) Common Crawl (public news) and (2) Alexandria (private news) are uncorrelated. Alexandria is a commercial dataset of sentiment scores based on news from subscription-based news sources. We conclude that Common Crawl can be seen as a *complementary* data source to private news sources (we state this clearly in section “Comparing the information transfer of private and public news to financial markets”). For a more in-depth comparison of the two sources, we refer to the Supplementary Discussion S1.

Additionally, in this study we did not consider the difference in latency between the availability of public and private news. In particular, in order to use public information in real-time, one has to scrape a large number of websites which adds a latency component. In our work, we do not use Common Crawl data in real-time mode, but in ex-post analysis with original news timestamps.

Moreover, we emphasize that commercially available sentiment data (such as that of Alexandria) may be trained/optimized for different time horizons than the daily horizon we consider here. In particular, Alexandria is designed for the purposes of *intra-day trading*.

Supplementary Figure S3. Correlation matrix of all simulated trading strategies using the Alexandria and Common Crawl News datasets. Each trading strategy has $n = 542$ daily returns.

Q7. A) A bit more detail on the profit generated. It is my understanding that a portfolio is rebalanced the next day with positions closed out. Were opening prices used?

Answer: We describe the trading strategy in detail in the first paragraph in section “Economical significance of news sentiment from Common Crawl Newsdata.” Indeed, we rebalance our portfolio positions daily at market open (9:30 AM). We use opening prices to calculate a portfolio’s performance. For additional information on the price data used, we refer to Supplementary Data S1.

Q7 B) I assume the positions are small so taking positions does not cause adverse moves? In reality there are always slippage costs. How would you account for this?

Answer: This is correct. Modeling realistic aspects of the market (e.g. slippage, transaction costs, market impact) is beyond the scope of our analysis. Our goal is not to create a profitable real-world trading strategy; rather, it is to analyze the information content of public news and its interaction with the financial market.

Q8. What about bid-ask spreads? Shouldn't such a cost be added to your position? For the Sharpe ratio, was regular calendar time or business calendar time used? *The last two questions are more from a finance perspective and so probably not as essential as the need to address the machine learning or NLP queries.*

Answer: Similar to Q7 above, we do not take bid-ask spread into account in our analysis. Yes, we use business calendar time.

Reviewer: 2

Comments to the Author(s)

This article takes a new stab at an old problem: How does news affect stock prices. It uses data from Common Crawl to extract finance-related news articles and extract news sentiments for stocks in S&P500. They then use transfer entropy to estimate how much the returns followed the news sentiment at the intra-day scale.

Finally, they pick the stocks most affected by the news and use the top 20 positive and negative ones to form a simple trading strategy. They show that this trading strategy outperforms both the ETF SPY and random strategies by a large margin.

I enjoyed reading this paper. While a great deal of work has been done on correlation of news sentiment with stock prices and using transfer entropy in stock markets, this way of combining the two and eventually verifying the predictions using a trading strategy seems novel to me.

If the authors can address my concerns, I recommend the paper for publication.

My main concern is the very high profit margin of your strategy.

As you may understand, profit of 21% is mind-blowingly high for day trading strategies. While I don't think it's impossible, usually even 2 sigma above random is quite an achievement. That's why I urge you to think about confounding factors. Could it be after-hour movements, encoded in news sentiments? Is it driven by the weird steady growth of the stock market from 2018-2020? What else can be confounding the result?

Answer: Our goal is not to create a profitable real-world trading strategy. Rather, it is to analyze the information content of public news and its interaction with the financial market. For this purpose, we use the trading strategy as an econometric instrument to measure the impact of news. The purpose here is to show that it has an impact. The simple trading strategy we use in our hypothetical simulation would most likely not be used "as is" as its turnover of 82.5% is high. To turn our economic instrument into a real-world trading strategy is much beyond the purpose and scope of our work.

Additionally, we emphasize that an annualized return of 21% is not that uncommon in hypothetical simulation experiments. For instance,

- Kelly et al. (2019), with their SESTM model, that uses proprietary data sources achieve a hypothetical annualized return of 33% (Sharpe ratio of 4.29); see Table 2 in their paper.

- Preis et al. (2013), achieve a hypothetical return of >300% over a seven year period based on “sentiment” derived from Google Trends (see Figure 2 in their paper).

Below, I have a few minor questions and comments.

Eq 2.2: what is the sum over? t , m and k ?

Answer: The sum goes over all elements in the domain of a joint distribution $p(r_{t+1}, r_t^{(m)}, s_t^{(k)})$. For notational simplicity, we omitted it. This notation is common practice in the literature of transfer entropy (e.g. Hlaváčková-Schindler, Katerina, et al. "Causality detection based on information-theoretic approaches in time series analysis." *Physics Reports* 441.1 (2007): 1-46] . In our particular case, we sum over the domain of the Cartesian product $S_1 \times S_1 \times S_2$. For S_1 and S_2 , we use symbolic re-coding for calculation of transfer entropy, by partitioning the data of returns and sentiments into a finite number of bins based on the quantiles (5%,25%,50%,75%,95%) of the empirical distribution. See Supplementary Discussion S2 for more details.

p6: MLE for \hat{p}_i : how difficult is the EM?

Answer: The EM has a closed form solution after approximating the set of sentiment-charged words S and the unobserved sentiments p_1, \dots, p_n . See Supplementary Methods S2 for details.

p7 line 54: why average sentiment, instead of sum?

Answer: When considering intra-day sentiment scores, we use hourly average sentiment scores in order to reduce the variance during different hours within a day. We do so to normalize. That said, it is likely that other transformations could also lead to statistically significant transfer entropies.

For each article you were summing word sentiments, so when binning, wouldn't it make more sense to sum within bins?

Answer: We do not sum word sentiments. The MLE estimate for \hat{p}_i on page 6 takes the sentiment p that maximizes the average sentiment. Note that we divide by s_i , i.e. the word count of the article. Summing word sentiments may significantly skew the distribution as longer articles will give rise to scores with different order of magnitudes.

p8: describing Day 1: there can be after-hour movements reported in the news, but showing up in the next day's prices. Can that be a significant contribution to the success of your strategy?

Answer: For the Day 1 portfolio, we use only past news for portfolio construction. Specifically, to compute our portfolio weights on day t we use news from 9:30 a.m. on day $t-1$ through 9:00 a.m. on day t . We then enter the market at market open (9:30 a.m.) on day t and hold our positions until 9:30 a.m. on day $t+1$. Hence, there is no look-ahead

bias in our Day 1 portfolio. In contrast, there is of course a look-ahead bias (as we point out in our article) in our Day 0 and Day -1 portfolios. In the Supplementary Discussion S1 we report on some additional hypothetical trading simulations to illustrate how the lookback window for the news impacts portfolio performance.

What other confounding factors can you think of?

Answer: In this study, we did not analyze possible confounding effects between multivariate time series of sentiment and stock price returns, as we were not focused on causal inference but on bivariate transfer of information between news to corresponding stocks. The multivariate case of transfer entropy can be further extended with partial transfer entropy in future work. We give appropriate pointers to this issue in our discussion.

Appendix B

Royal Society Open Science
publishing@royalsociety.org

Dear Editor,

Thank you you're the acceptance of manuscript RSOS-202321.R1 "On the impact of publicly available news and information transfer to financial markets" has been accepted for publication in Royal Society Open Science subject to minor revision.

We have made appropriate minor revision changes and also submit the clean verison.

Please find the response to referees, below:

Reviewer comments to Author:
Reviewer: 1

Comments to the Author(s)

Thank you. I urge the authors to emphasize their paper is not advocating a profitable trading strategy but rather the interplay between public information and the markets. This link is tenuous and can be better explained or motivated.

>> Thanks, we have made the following change that is denoted with red color.

By using a simple sentiment-based trading strategy as an econometric tool only and not with the intention of being a realistic and implementable trading strategy, we find that our sentiment signals from public news are carrying both economic value and complementary information compared to non-public and commercial news data providers.

Reviewer: 2

Comments to the Author(s)

I am satisfied with the response and recommend the paper for publication.

One clarification regarding my question on summing sentiments: I meant summing per article sentiments. Each article's sentiment of course needs to be normalized by the word count. But the mean sentiment makes sense. I think the look-ahead effect is also excluded properly in your method and the clarification added in the paper regarding bivariate correlation and not causal effects seems adequate.

>> Thanks for clarification and for your recommendation of acceptance.

We are confident that the revised manuscript is now suitable for publication in the Royal Society Open Science journal.

Yours faithfully,

Metod Jazbec, Barna Pasztor, Felix Faltings, Nino Antulov-Fantulin and Petter N. Kolm

Corresponding authors

Dr. Nino Antulov-Fantulin

Computational Social Science
Stampfenbachstrasse 48
STD Building, F Floor
8092 Zürich, Switzerland
anino@ethz.ch

Dr. Petter N. Kolm

NYU Courant
251 Mercer St., Room 520
New York, New York 10012
212-998-4855
petter.kolm@nyu.edu

ETH zürich

NYU

**COURANT INSTITUTE OF
MATHEMATICAL SCIENCES**